# Mechanics of 3D-Printed Polymer Lattices with Varied Design and Processing Strategies

**DOI:** 10.3390/polym14245515

**Published:** 2022-12-16

**Authors:** Paul F. Egan, Nava Raj Khatri, Manasi Anil Parab, Amit M. E. Arefin

**Affiliations:** Mechanical Engineering, Texas Tech University, Lubbock, TX 79409, USA

**Keywords:** 3D printing, resin, lattices, design, mechanics, dimensional characterization

## Abstract

Emerging polymer 3D-printing technologies are enabling the design and fabrication of mechanically efficient lattice structures with intricate microscale structures. During fabrication, manufacturing inconsistencies can affect mechanical efficiency, thereby driving a need to investigate how design and processing strategies influence outcomes. Here, mechanical testing is conducted for 3D-printed lattice structures while altering topology, relative density, and exposure time per layer using digital light processing (DLP). Experiments compared a Cube topology with 800 µm beams and Body-Centered Cube (BCC) topologies with 500 or 800 µm beams, all designed with 40% relative density. Cube lattices had the lowest mean measured relative density of ~42%, while the 500 µm BCC lattice had the highest relative density of ~55%. Elastic modulus, yield strength, and ultimate strength had a positive correlation with measured relative density when considering measurement distributions for thirty samples of each design. BCC lattices designed with 50%, 40%, and 30% relative densities were then fabricated with exposure-per-layer times of 1500 and 1750 ms. Increasing exposure time per layer resulted in higher scaling of mechanical properties to relative density compared to design alteration strategies. These results reveal how design and fabrication strategies affect mechanical performance of lattices suitable for diverse engineering applications.

## 1. Introduction

Emerging 3D-printing technologies are enabling the design and fabrication of complex, mechanically efficient lattice structures from polymer materials [1,2,3]. 3D-printed lattice structures are promising for a diverse range of engineering applications, including biomedical tissue scaffolds, safety helmets, and personal protection equipment [4,5,6]. Lattice structures are built from repeating substructures that form intricate architectures and necessitate consideration of numerous mechanical and dimensional trade-offs [7,8,9]. Recent successes in 3D-printed lattice design have used beam-based approaches with unit cells that form efficient structural units, such as trusses, that are repeated to form a complete lattice with tuned mechanical capabilities [10,11]. However, manufacturing constraints and fabrication inconsistences limit the performance achieved by printed structures [12,13,14], thereby necessitating new scientific approaches to understand how manufacturing variations affect mechanical performance. Here, experimental investigations are conducted to determine how lattice design and processing affects lattice dimensional accuracy and mechanics, with a focus on testing large numbers of samples to better understand distributions and variances in 3D-printed lattice properties.

There are numerous 3D-printing processes capable of producing lattice structures, including extrusion, resin, and powder-based printing [15,16]. Each process has relative advantages and disadvantages. Fused deposition modeling and fused filament fabrication are relatively inexpensive extrusion processes with large material libraries that have generally poor resolution and consistency for fabrication at sub-millimeter scales [17,18]. Powder printing processes, such as selective laser sintering or selective laser melting, have more consistent sub-millimeter resolution but are relatively expensive and generally limited to nylon and metal materials [19,20]. Resin-based processes, such as stereolithography or digital light processing (DLP), provide a strong middle ground for balancing functionality with pricing, with a wide range of polymers available for printing that can achieve accuracy on the order of a few hundred microns or lower [21,22,23]. DLP printing operates by projecting light to cure a vat of resin to form an entire layer of a printed structure at once, which is an efficient build process with comparable or superior resolution to fused deposition modeling, liquid crystal display, and stereolithography printing [24,25,26]. Although DLP printing is highly accurate, there are still microscale variances even for simple lattices with Cube unit cells with beam diameters of 0.11 to 1.05 mm that require corrections to ensure printing uniformity [27]. Corrections are possible by altering beam diameters or exposure time per layer, but further experimental studies are necessary to determine how corrections affect mechanical outcomes.

Although many types of lattices have been experimentally and theoretically characterized [28,29,30], it is not straight-forward to predict how manufacturing inconsistencies introduced by 3D-printing processes affect mechanical outcomes [31,32]. Key mechanical properties of lattices, including elastic modulus and yield strength, are highly dependent on lattice design configuration and printing processes [33,34]. For instance, in PolyJet printing, which is also a resin-based printing process, inconsistency in beam diameters was found to influence mechanics with a topology dependence. Lattices with Cube unit cells were more likely to fail in printing compared to Body-Centered Cube (BCC) unit cells [35,36]. In fused deposition modeling, the large number of possible factors in processing necessitates extensive design of experiments to determine which processing strategies result in optimal prints [37,38]. In DLP printing of polyurethane acrylate, experimental results were sensitive to strain rates and anisotropy, and structures demonstrated complex fracture behaviors depending on loading conditions [39]. Understanding how design and fabrication affect mechanical performance is essential for designing structures for specified applications, where DLP printing provides unique capabilities for applications such as bone tissue engineering [40,41]. Bone tissue scaffolds have been previously investigated using stereolithography printing for altering lattice topological design [42]. However, there remains limited understanding regarding how strategies such as beam diameter resizing or process parameter alterations may affect resulting mechanics. DLP printing provides potential improvement for fabricating prints that mimic bone’s microstructure and mechanics, while also enabling multiple new design and processing strategies for tuning lattice structure and mechanics.

Therefore, this study focuses on the mechanical and dimensional characterization of polymer lattices fabricated with varied design and processing strategies using DLP printing. Design alterations include topology, beam diameter, and unit cell length to create lattices of different relative densities within the constraints of DLP polymer printing technologies. Cube and BCC unit cells were selected to form lattices with 40% relative density for benchmarking statistical distributions of print dimensions and mechanics for 30 samples each. Mechanical characterization includes unconfined compression testing to determine elastic modulus, yield strength, and ultimate strength to benchmark lattice performance. A second experiment was conducted by varying lattice’s relative density from 50% to 30% with 1500 and 1750 ms layer exposure times to compare how different design and processing strategies affect lattice manufacturing and mechanics. The study provides novelty through comparing design and processing strategies for polymer 3D printing in relation to actual measured outcomes that may differ from ideal cases for a variety of conditions. Outcomes are expected to highlight which design and printing strategies are most advantageous for maximizing performance of lattice structures for specified applications.

## 2. Materials and Methods

### 2.1. Lattice Design

Lattices were designed using a Python script to automate lattice construction from beams organized as unit cells using macros with Abaqus software (Dassault System, Waltham, MA, USA, version 6.19-1) [43]. Beams were patterned as unit cells with specified topologies and relative density of the lattice calculated based on the ratio of the structural volume to nominal volume. Unit cells were constructed with a cubic topology that consisted of a Cube unit cell with beams along each cubic volume edge and a BCC unit cell that consisted of beams along each cubic volume edge with additional beams from each corner to the volumetric center. Unit cells were designed with beam diameters of 500 µm or 800 µm. All beams within a lattice had homogeneous diameters. Unit cell length was then increased until the overall relative density of a lattice with six-unit cells along each axis reached a target relative density. The use of six-unit cells reduces boundary layer effects that are sources of error for mechanical testing. Three lattices were designed for mechanical and dimensional characterization, as demonstrated in Figure 1, with relative densities of 40%.

The first lattice had a Cube unit cell, 800 µm beam diameter, and 1.8 mm unit cell length and is referred to as the Cube-800ø design. The second lattice had a BCC unit cell, 800 µm beam diameter, and 3.1 mm unit cell length and is referred to as the BCC-800ø design. The last lattice had a BCC unit cell, 500 µm beam diameter, and unit cell length of 1.9 mm and is referred to as the BCC-500ø design. The design of these three lattices provides a basis for comparison with one design parameter controlled each in relation to the BCC-800ø design, which is only different from the Cube-800ø design by unit cell topology and the BCC-500ø design based on its beam diameter design. Designs of lattices were determined based on constraints in the build process such that lattice beam diameters were large enough for printing based on native resolution of the printing system, while ensuring lattices were small enough for printing an entire batch of thirty lattices at once.

### 2.2. Build Process

Lattices were fabricated using a DLP printing approach with an Envision One cDLM printer (EnvisionTEC, USA) using E-RigidForm Amber resin (EnvisionTec, Dearborn, MI, USA) that has an elastic modulus of approximately 3000 MPa. The Envision One printing system has a native xy-resolution of 93 µm, z-resolution of 50 to 150 µm, an ultraviolet light-emitting diode (UV-LED) light source at 385 nm wavelength, and build envelope of 180 by 101 by 175 mm. Prints commenced with manufacturer layer thickness of 50 µm, support thickness of 0.8 mm, support spacing of 1.75 mm, upright orientation, ambient temperature, initial burn-in time of 17,500 ms, waiting time before exposure of 1000 ms, separation velocity of 300 µm per second, and exposure time of 1500 ms or 1750 ms per layer. Support material size was determined through preliminary testing to minimize support material to ensure its removal without destroying the surrounding structure while ensuring support material was printed large enough to facilitate structural stability during printing. Preliminary work and past studies informed these parameters as suitable limits for high-quality prints according to hardware constraints [41].

Envision One Rapid Prototype software (EnvisionTec, Dearborn, MI, USA, version: 1.23.4820) was used to prepare digital files from STLs and add support material structure. All samples were printed with an orientation indicated by Figure 2, where build layers are formed sequentially in the build direction ($\vec{b_{dir}}$) by projecting light for a specified period of time to cure one layer of resin, as lattices are built in a layer-by-layer fashion.

Samples were removed from the build platform using a metal spatula, rinsed by hand with isopropyl alcohol, then placed in a bath of isopropyl alcohol with stirring to remove excess uncured resin. Support material was removed from each sample using a metal blade. Samples were placed in a PCA 2000 ultraviolet curing chamber (EnvisionTec) for two-hours for postprocessing. Thirty samples were fabricated for each of the Figure 1 designs using an exposure time per layer of 1750 ms.

### 2.3. Dimensional Characterization

The height H and length L of lattices were measured with calipers, with height referring to the dimensional axis aligned with the build direction. The relative density is defined as the ratio of solid material volume to nominal volume of the lattice. The weight w of lattices was also measured to determine the lattice relative density ($\rho_{r}$) through comparison with the solid material density ($\rho_{m}$), measured as 1.19 g/cm^3^, and input for Equation (1) [44]:
(1)$$\rho_{r}=\frac{w}{\rho_{m}\cdot H\cdot L^{2}}$$

An Olympus DSX-HRSU digital microscope (Olympus Corporation, Tokyo, Japan) with 5× objective was used to produce stitched images of lattice faces to enable measurement of beam diameters. Images were stacked to produce an in-focus image that accounted for the depth of diagonal beams being at a different focal plane than orthogonal beams. Beam diameters were measured for unit cells on the side face of lattices, where side faces refer to those that are perpendicular to the build platform during fabrication. Mean beam diameters were measured for three representative samples of each design and categorized according to the beam’s orientation relative to the build direction. Figure 3 demonstrates the stitched microscope image of a complete lattice face, with markings demonstrating how individual beam measurements were collected for each beam orientation.

Beams that were aligned with the build direction were labelled as 0°, those that were diagonal (for BCC topologies only) were labelled as 45°, and those that were perpendicular to the build direction were labelled as 90°. For each lattice face, the unit cells making up the two middle columns of the face were used to collect measurements, as indicated in Figure 3, for two 0°, two 45°, and two 90° measurements for each unit cell. Due to the overlapping of unit cells, the total measurements for one lattice face resulted in fourteen 0° beam diameter measurements, twenty-four 45° beam diameter measurements, and eighteen 90° beam diameter measurements.

### 2.4. Mechanical Testing

Mechanical testing was conducted using an Instron 5966 universal testing machine (Instron, Norwood, MA, USA) and protocols according to the ISO-13314 standard [45] that are consistent with previous studies for 3D-printed lattices [7]. Lattices were placed between the compression plates, as demonstrated in Figure 4, such that the top/bottom faces were in contact with the plates while side faces had no contact with the plates. Testing was conducted with a loading rate of approximately 0.1 strain per minute, with tests concluding once 0.25 strain was reached, which ensured all samples were tested past yielding.

Mechanical testing results were analyzed using a Python script to calculate the mechanical properties of lattices, including elastic modulus, yield stress, and ultimate stress. Elastic modulus ($E_{m}$) was calculated from measured stress-strain curves by determining a lower strain bound ($\varepsilon_{1}$) and an upper strain bound ($\varepsilon_{2}$) in the linear-elastic region, with corresponding lower ($\sigma_{1}$) and upper stress ($\sigma_{2}$) values, according to Equation (2) [45]:(2)$$E_{m}=\frac{\sigma_{2}-\sigma_{1}}{\varepsilon_{2}-\varepsilon_{1}}$$

The yield strength ($Y_{s}$) was determined using the 0.2% offset approach. The ultimate strength ($U_{s}$) was determined as the highest stress measured for each curve. Mechanical property measurements were tracked individually for each sample to couple measurements with the relative density measurement described in Section 2.3, which allows for plotting the measured relative density against the measured mechanical properties across samples.

### 2.5. Design and Process Strategies

A second set of three lattices were designed by varying relative density while retaining the BCC topology, using the BCC-500ø as a control. The set of three BCC topology designs all had 500 µm beam diameters with unit cell lengths of 1.6 mm, 1.9 mm, and 2.3 mm to create designs of 50%, 40%, and 30% relative density (Figure 5). These designs were printed at an exposure time per layer of 1500 and 1750 ms. Mechanical testing of these designs enables comparison of strategies for altering designs or processing times to determine which approach leads to more favorable printed structures and mechanical properties. Five replicates of each design were printed and tested for each processing condition.

## 3. Results and Discussion

### 3.1. Fabricated Designs

Thirty samples of each Figure 1 lattice design were fabricated and measured for dimensional characterization of lattice height, lattice length, and relative density, with mean measurements indicated in Table 1. The lattice length is calculated based on all lattices having six-unit cells along each axis and adjacent unit cells sharing beams. Height and length are not differentiated in the design due to symmetry. However, height and length are differentiated in measurements because of fabrication dependency of the build direction. The designed relative density for all samples was approximately 0.4.

Measurements generally indicate strong dimensional accuracy for lattice height and length for all designs, with no more than a 0.2 mm difference on average when considering all comparisons. The relative density was most accurate for the Cube-800ø design (0.42), then the BCC-800ø design (0.46), and least accurate for the BCC-500ø design (0.54). The differences in relative density accuracy may occur due to the smaller spacing between beams in the BCC-500ø design compared to the other two designs, which leads to a high accumulation of resin curing in void regions of the lattice. These findings are in agreement with past studies for resin printing that have demonstrated greater resin accumulation and fusion toward the center of lattice structures [27]. The resin accumulation potentially occurs due to inner pores having impeded flow for resin due to further distance from lattice boundaries. This accumulation is demonstrated on the top surface of the BCC-500ø design, as indicated in Figure 6.

The fabricated designs in Figure 6 demonstrate that the overall size of the BCC-800ø sample is larger than the Cube-800ø and BCC-500ø designs based on its larger unit cell height. Some further defects demonstrated are that the bottom part of the Cube-800ø design has a waviness, and the beams along the top of the BCC-500ø structure appear curved. Differences in achieved lattice measurements are further demonstrated in Figure 7, where the measured relative density for each sample is plotted on a histogram, using intervals of 0.01 for bin sizes.

Figure 7 demonstrates the frequency distribution of relative density measurements such that the Cube-800ø design has the lowest range of relative density measurements of about 0.40 to 0.43, while the BCC-500ø samples range from 0.46 to 0.59, and the BCC-800ø samples are in between, with a range of 0.42 to 0.50. The smaller range in relative densities for the Cube-800ø design led to a higher frequency of lattices with the same relative density, with 11 samples having a relative density of 0.42, which demonstrates a higher consistency in fabrication than the other designs. The distributions of Cube-800ø and BCC-800ø designs followed a more normal distribution than the BCC-500ø design, which had a fifth of its samples with 0.5 relative density, while approximately half of the samples had 0.54 to 0.57 relative density. Further differentiation in why relative density varied among lattices is possible to investigate through microscale dimensional characterization and observation.

### 3.2. Microscopic Characterization

The microstructures of lattices were imaged for three samples of each design, with representative images provided in Figure 8. The images captured a 6 × 6 array of unit cells across the entire face of each lattice. Microscopy measurements indicate a regular patterned structure for all lattices. There is less consistency in pore sizes for BCC lattices, especially for the BCC-500ø lattice, where more cured resin is accumulated toward the center, which leads to smaller pores.

Beam diameters were measured and aggregated for the mid-point of beams on the vertical column of unit cells for each microscope image, which resulted in the measurement of forty-two 0° beam diameter measurements and fifty-four 90° beam diameter measurements for each design, with seventy-two 45° beam diameter measurements for each BCC design, according to methods described in Section 2.3. Mean measurements were determined for each design and plotted in Figure 9.

Figure 9 demonstrates the dependence of mean beam diameter on the beam’s orientation relative to the build direction, with 0° beams measured higher than the beam’s intended diameter for all designs, while 90° beams were smaller for Cube-800ø and BCC-800ø designs. 45° beams were, on average, larger than intended for both BCC designs. When aggregating results for all beams of all orientations, the mean beam diameter of the Cube-800ø design is ~800 µm, ~810 µm for the BCC-800ø design, and ~550 µm for the BCC-500ø design, which contributes to its higher relative density than other lattices. Although these mean measurements are close to the design’s intended value for 800 µm beams, individual beams and orientations can still be quite different from the mean due to manufacturing variations. The BCC-500ø design had a measured mean beam diameter greater than intended for all orientations, which is possibly due to higher resin accumulation around its smaller pores and/or errors in printing resolution having a greater proportional effect for these smaller beams. Comparable variations for 50% relative density lattices were observed in PolyJet printing, where beams intended for 500 µm diameters ranged from 280 to 510 µm and from 770 to 1100 µm for 800 µm designs [7]. In contrast, the ranges for diameters measured for Figure 9 with DLP printing were 400 µm to 700 µm for 500 µm designs and 620 µm to 960 µm for 800 µm designs. These findings show that DLP printing had a much higher minimum diameter for the 500 µm design when comparing to similar structures from PolyJet printing. These larger minimum diameters mitigate the possibility of failures that may occur due to localized weak points in the lattice.

### 3.3. Lattice Mechanics

The thirty fabricated samples for each Figure 1 design were mechanically tested in compression, with stress-strain curves plotted in Figure 10 for each sample. Each design had a linear elastic region followed by yielding. After yielding, the behavior differed for each design. Stress increased for the Cube-800ø and BCC-500ø designs, while it stayed consistent for the BCC-800ø designs. In cases where stress increases after yielding, densification of the structure may play a role, whereas bending of individual beams may play a larger role in the failure mechanisms of the BCC-800ø designs compared to the Cube-800ø and BCC-500ø designs with smaller unit cells lengths. These differences in behavior may occur due to the differences in relative length scales between beams and unit cell sizes, with longer, more slender beams being more likely to buckle. Additionally, the greater relative density of the BCC-500ø design also promotes a greater likelihood of densification since the fabricated lattice has a structure closer to a solid material compared to the 800 µm beam diameter designs.

The mean mechanical properties for elastic modulus, yield strength, and ultimate strength are plotted for each design in Figure 11. For mean elastic modulus, the Cube-800ø (~310 MPa) and BCC-500ø (~330 MPa) have higher values than the BCC-800ø design (~270 MPa), which is a trend that continues for the yield strength measurements. These differences occur because the Cube-800ø design has a higher proportion of its beams aligned with the loading direction, while the BCC-500ø design’s higher mechanical properties are attributable to its higher relative density after fabrication. These mechanical variations are in agreement with simulation results that suggest higher relative density designs will have higher mechanical properties and that directional properties are dependent on the alignment of beams throughout the lattice [46]. The BCC-500ø design had a much higher mean ultimate strength than the other two designs due to its deformation that led to a continuously increasing stress applied per strain, according to Figure 10 results, that might indicate the beginning of densification.

The mechanical properties of lattices were then investigated by considering the coupled measurement between each sample’s relative density and resulting mechanics to determine how the variation in relative density due to fabrication inconsistencies affects mechanics. Mechanical properties were fit to a linear regression model to determine correlations, according to Equation (3) [7]:
(3)$$p^{*}=m\cdot \rho_{r}+c$$

In Equation (3), $p^{*}$ is interchanged with the appropriate mechanical property of elastic modulus ($E_{m}$), yield strength ($Y_{s}$), or ultimate strength ($U_{s}$), while m represents the slope of the linear regression, $\rho_{r}$ represents the relative density, and c is the offset. The physical meaning of Equation (3) is if m is higher, then mechanical properties scale at a higher rate with relative density which implies a higher mechanical efficiency. Physically, the mechanical efficiency is dependent on the placement of material in lattice. Increases in relative density that result in higher mechanics, such as increasing beam diameters, are more likely to increase mechanical efficiency than random distributions of extra material from fabrication errors. Results for each regression are presented in Table 2.

Regression results demonstrate a positive slope for all relationships, which indicates that an increase in relative density results in an increase in mechanical properties. The magnitude differs across designs, with a generally steeper slope for BCC-800ø and BCC-500ø mechanical properties. The correlational coefficient is generally higher for BCC designs than the Cube design, thereby indicating less expected variation when considering how increases in relative density will affect the targeted properties. These relationships are all plotted with individual measured data points for each sample in Figure 12. 

Figure 12 demonstrates the range of mechanical properties and relative densities achieved by each design, which illustrates the larger range of relative densities of the BCC designs compared to the Cube design. The comparatively higher correlation coefficient values of the BCC-800ø designs are also demonstrated with data points being more tightly coupled to the regression line. A reason for this difference between the BCC-800ø and BCC-500ø designs may be that the extra material added in the BCC-500ø structures does not always contribute to better mechanics if it is added in a non-structurally efficient manner, such as solid regions and closed pores throughout the lattice. The larger spacing between beams for the BCC-800ø design may allow for added material to contribute directly to improved mechanics through an efficient placement that bolsters beam diameters and joints effectively. Likewise, the Cube lattice may not have an efficient placement of extra material due to variations from the fabrication process because many beams are not aligned with the loading direction, thereby limiting the extra material’s usefulness.

Figure 12 highlights novel results when comparing large numbers of lattices printed under controlled conditions, whereas commonly conducted research focuses on differences between small batches of printed structures with different design features. When considering an ideal printing case, all samples of a given design in Figure 12 should have the same relative density, but manufacturing variations demonstrate a variance in relative density that is directly proportional to mechanical properties. Each lattice design has a unique relationship, according to Table 2, for how relative density scales with mechanics, which suggests that design plays a role in fabrication accuracy in addition to manufacturing processes. Next, experiments were conducted for altering lattice density through rescaling lattice unit cell size and altering exposure time per layer to determine how these strategies affect mechanical outcomes.

### 3.4. Process Effects

BCC topologies were designed with 500 µm beam diameters and relative densities of 50%, 40%, and 30% (Figure 5) to determine how controlled changes in relative density for lattice designs scales with mechanics compared to those of manufacturing variations. Processing effects from exposure time per layer were set at 1500 or 1750 ms to determine how altering achieved relative density through processing variations affects manufacturing accuracy and resulting mechanics. These designs were selected because the BCC-500ø designs demonstrated the greatest inaccuracies in measured relative density and mechanics compared to the 800 µm beam diameter lattices investigated in Section 3.3, and thereby demonstrate the greatest potential for improvements through altered processing. Results are presented in Table 3 for five samples tested for each condition.

Table 3 results demonstrate that designs printed with 1500 ms exposure times have more accurate relative densities than those of 1750 ms, which are generally too high. For both exposure times, accuracy was higher for lattices with lower relative density, which is likely due to less resin accumulation due to the larger pores in these lattices. Mechanical properties increased with higher exposure time and higher relative density designs. These results are in agreement with similar resin printing processes that have demonstrated resin accumulation that is dependent on the size and location of pores throughout a lattice, with greater fusion occurring toward the center [27]. Fabricated lattices are demonstrated in Figure 13, which indicates the larger proportion of fused pores for higher exposure designs with lower relative density.

Fabrication accuracy was further assessed using microscopy to determine consistency of beam diameters and pores on lattice surfaces (Figure 14). Results demonstrate that pores were open throughout all structures. There is less consistency in lattices with higher exposure time and lower relative density. Inconsistencies include fusion of pores and irregular patterning due to unit cells, which is noted in the 1500 ms BCC-0.5$\rho_{r}$ design, where the top surface is deformed, with highly rounded edges.

The mean beam diameters for each design and exposure time were measured according to Section 2.3 methods and demonstrated that higher exposure time increased mean beam diameters for each design (Figure 15). For instance, 90° mean beam diameters for 70% relative density designs were 420 µm for 1750 ms exposure time per layer and 320 µm for 1500 ms conditions. All printed beam diameters were, on average, lower than expected. When considered in tandem with Table 3 results that demonstrate that relative densities were generally higher than printed, these results suggest that larger beam diameters and fusion may occur more toward the center of the lattice since surface-level beam diameters were measured smaller than expected. These differences are similar to measurements for different angles of printing that have been documented in prior DLP printing research, where part angulation plays a significant role in accuracy [26]. These findings suggest that lattices can be redesigned with different proportions of beams or sizes of beams based on their orientation to ensure greater consistency for the overall print.

When comparing designs of different relative densities, the beam diameters remained consistent. For instance, for the 50%, 40%, and 30% relative density designs of 1750 ms exposure time per layer, the 0° mean beam diameters were 480 µm, 460 µm, and 480 µm, respectively. These results suggest that exposure timer per layer has a greater effect on mean beam diameter accuracy compared to altering unit cell length.

Mechanical results for elastic modulus, yield stress, and ultimate strength were plotted when considering all fifteen samples printed and tested for a given layer exposure time as a single data set (Figure 16). Clusters of data around particular relative density values are representative of individual measurements for each of the five samples designed at 50%, 40%, and 30% relative densities. Generally, higher exposure times led to increased mechanical properties per density for elastic modulus and yield strength, with a minimal increase for ultimate strength compared to lower exposure time conditions. The higher exposure times improve mechanics through increasing beam sizes (and therefore relative density). The reason ultimate strength may not scale as strongly with exposure time is due to it being measured as the lattice is beginning densification. Densification occurs as the material begins to behave more as a solid such that there are fewer dependencies of behavior based on initial topological variations.

All data was fit with regression models according to Equation (3) when considering the relative density of all printed designs based on the set exposure time to quantify how relative density scales with mechanical properties for each condition (Table 4). Each regression represents the fitting of a line to fifteen measured relative densities for each exposure time (i.e., five measured relative densities for 50%, 40%, and 30% relative density designs, as demonstrated in Figure 16).

Regression results demonstrate a positive correlation with relative density for each mechanical property for all designs, which is illustrated with Figure 16 results. There is a higher slope of lines for 1750 ms designs that suggest the higher exposure time leads to better improvement of mechanics with relative density. When considering the designs of a similar measured relative density for different exposure layer times, the measured mechanics of 1750 ms designs were higher than those of 1500 ms designs. These results show that even though lattices of the same topology have similar relative density, there is a difference in performance based on design and processing strategies.

Overall, these results demonstrate that lattices of the same relative density can have different scaling with mechanical properties depending on each lattice’s design configuration and process parameters. Previous studies have demonstrated such phenomena for printing processes, such as fused deposition modeling [37,47], but less research has investigated process effects on mechanics of resin-printed parts, especially when considering property variations from bulk printing. At measured relative densities of around 30%, there is not much difference in mechanical properties based on exposure time. At higher relative densities, lattices with higher exposure times have increased mechanical performance per density compared to lattices that had increased relative density due to a smaller designed unit cell size. These results demonstrate how variations in a lattice’s relative density due to manufacturing affects mechanical performance, which informs future engineering design strategies to process polymers for 3D printing to maximize performance for specified applications.

### 3.5. Further Discussion

Mechanical and dimensional characterization was conducted for sets of lattices with different topology, beam diameter, and relative density designs in addition to altered processing by varying exposure time per layer using DLP printing. Printing demonstrated that lattices were qualitatively printed accurately with a distribution of measured relative densities that primarily varied due to resin accumulation around pores. In comparison to fused deposition modeling, PolyJet, and sintered lattices, the DLP printed lattices generally have a high fidelity in reproducing microscale features. For instance, fused deposition modeling/fused filament fabrication beams tend to have globules of material that create large discrepancies in diameters [48,49], PolyJet lattices tend to have sharp irregularities in layers of beams at the sub-millimeter scale [7], while sintering processes have irregularities based on groupings of powder particles that fuse together [45]. These differences in microstructures for printing processes lead to different mechanical behaviors. For instance, the sharp irregularities of PolyJet lattices lead to weak points for fractures, while DLP printed lattices have more uniform beam diameters for consistent mechanics.

Mechanics were investigated for differently designed lattices for two sets of experiments. Both sets of experiments contained a control design referred to as BCC-500ø or BCC-0.4$\rho_{r}$. Having a control design enables comparisons for how effects of topology, relative density adjustment, and processing all affect lattice manufacturability and mechanics by considering one variable at a time. Topology differences showed that BCC structures have greater manufacturing variability than Cube topology structures and lower mechanical properties per density. Larger beam diameters were printed more consistently, thereby leading to less mechanical variation in performance. Increased relative density design and exposure time per layer both improved mechanics, although increasing the exposure time per layer provided a higher increase in mechanical properties per relative density, thereby suggesting that it is a more mechanically efficient design approach. By considering all of these results together, the most mechanically efficient and consistent design for uniaxial compression would be a Cube topology with 800 µm beam diameters printed with an exposure per layer time of 1750 ms. However, when considering shear loading cases where diagonal beams provide benefits over orthogonal beams [50], the BCC topologies could provide a more balanced mechanical response.

When comparing the BCC-500ø and BCC-0.4$\rho_{r}$ design across experiments, the lattices’ measured properties should remain constant, but differences were observed. The average relative density was 0.54 and 0.51, and elastic modulus was 330 MPa and 350 MPa, respectively. These differences in measurements highlight stochasticity in the printing process across batches, which could be due to differing placement of lattices on the build platform between the 30 samples printed in the first experiment and five samples printed for the second experiment, different mixing/state of the resin, and further confounding factors. These differences highlight the importance in controlling printing processes as much as possible to ensure consistent curing of polymers to form structures. Additionally, these results confirm the need to consider large numbers of samples when analyzing experiments due to the inherent variation of 3D-printed structures. The thirty samples printed in the first batch demonstrated a large variance in relative density, from 0.46 to 0.59, that is unlikely to be captured through typical experiments in the field that consider five or less printed and tested samples. However, the small batch size of the second experiments did provide similar mean values to the larger batch, which suggests that the approach is valid to broadly test design and processing strategies prior to committing more resources toward bulk printing for more rigorous statistical analysis.

Limitations exist from the extensive time required for experiments, imaging techniques, and mechanical characterization that could be investigated further in future investigations. Future work could investigate further trade-offs in topology, relative density, and beam diameter mechanics using design for experiments, in particular, by conducting experiments to determine how mechanical efficiency scales with varied topology, design, and processing strategies. Further processing experiments could consider higher or lower exposure times. However, lower exposure time per layer may result in no printed structure forming, while higher exposure leads to further fusion of beams that further deviate from the mechanical efficiency provided by lattice structures. Microscopy imaging only considered one face of each lattice at a time to infer the inner structure density based on relative density measurements, but future studies could investigate beams throughout the inner volume of lattices, which is possible with microCT analyses [36,51]. Further mechanical testing could link dimensional characterization to failure modes of lattices and mechanical performance for scenarios such as energy absorption [52,53], which is commonly considered across engineering applications. Further analysis of mechanical variations for different design and processing conditions could lead to a more complete understanding of how fabricated designs differ in performance from theoretical models. Such understanding may promote new design and manufacturing approaches to improve lattice mechanical performance across applications.

## 4. Conclusions

This study investigated the mechanics of lattices configured with two different topologies (Cube/BCC), three different designed relative densities (50%, 40%, and 30%), and two different exposure times per layer (1500 and 1750 ms) using DLP printing. Initially, thirty samples each of Cube and BCC unit cell topologies with beam diameters of 500 µm or 800 µm were printed with 40% relative density. The mean measured relative density was higher than intended for all designs and ranged from 42% for the Cube-800ø design to 54% for the BCC-500ø design. These higher relative densities are attributed to regions of solid fusions around pores, especially in the smaller spacing of the BCC-500ø design. Elastic modulus, yield strength, and ultimate strength were tested and plotted against relative density for each sample, which demonstrated a positive correlation between mechanical properties and relative density due to manufacturing deviations for all designs. A second experiment was conducted to compare strategies for redesigning BCC-500ø designs with 50% or 30% relative density and altering the exposure time per layer. When comparing printed designs of similar relative density measurements, those that were processed at higher exposure times had greater mechanical efficiency (i.e., higher mechanical properties per relative density).

Key conclusions of the study are: (1) manufacturing variations in lattices affect relative density distributions; (2) higher relative density from manufacturing variations results in higher relative mechanical properties; (3) scaling of mechanical properties with relative density varies based on design and processing strategies; and (4) increasing exposure time per layer results in greater improvements in mechanical efficiency than investigated design alteration strategies. The findings highlight a need for considering and evaluating diverse polymer processing strategies to promote high mechanical efficiency of 3D-printed polymer lattices. Future work could further investigate how topology, materials, and processing each uniquely affect manufacturing and mechanics to find optimal strategies for lattice construction that benefit diverse engineering applications.

## Figures and Tables

**Figure 1 polymers-14-05515-f001:**
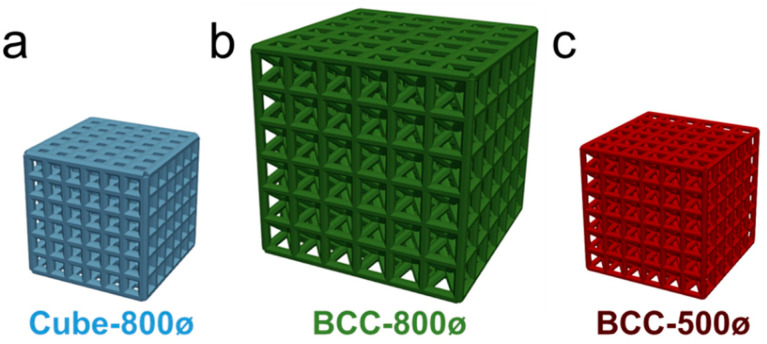
Designs of 40% relative density for (**a**) Cube-800ø lattice with Cube unit cells with 800 µm beam diameters; (**b**) BCC-800ø lattice with BCC unit cells with 800 µm beam diameters; and (**c**) BCC-500ø lattice with BCC unit cells with 500 µm beam diameters.

**Figure 2 polymers-14-05515-f002:**
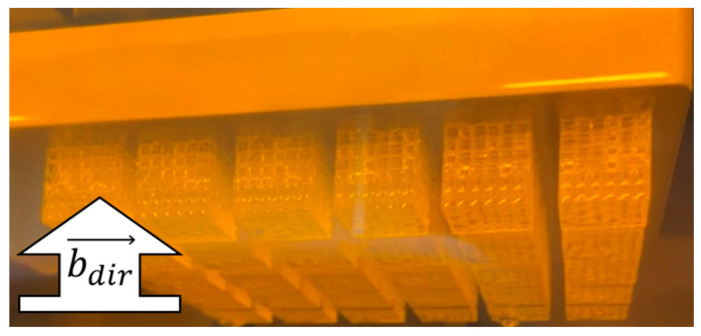
Fabricated lattices on build platform.

**Figure 3 polymers-14-05515-f003:**
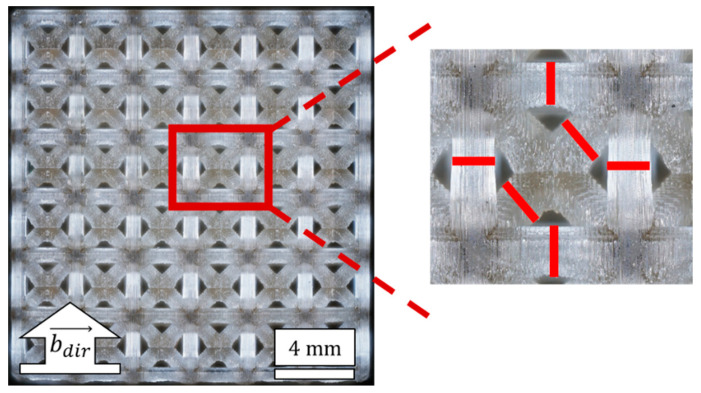
Microscopy of lattice side face and demonstrated measurements for beam diameters aligned (0°), diagonal (45°), and perpendicular (90°) to the build direction.

**Figure 4 polymers-14-05515-f004:**
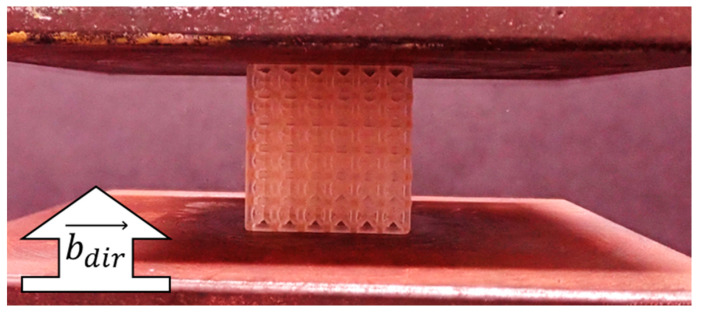
Mechanical testing for lattice compression.

**Figure 5 polymers-14-05515-f005:**
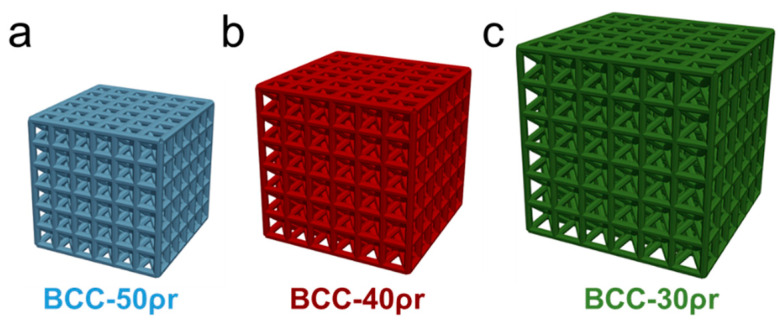
Designs for (**a**) BCC-50$\rho_{r}$ (**b**) BCC-40$\rho_{r}$, and (**c**) BCC-30$\rho_{r}$ lattices designed with BCC unit cells of 500 µm diameter with 50%, 40%, and 30% relative densities, respectively.

**Figure 6 polymers-14-05515-f006:**
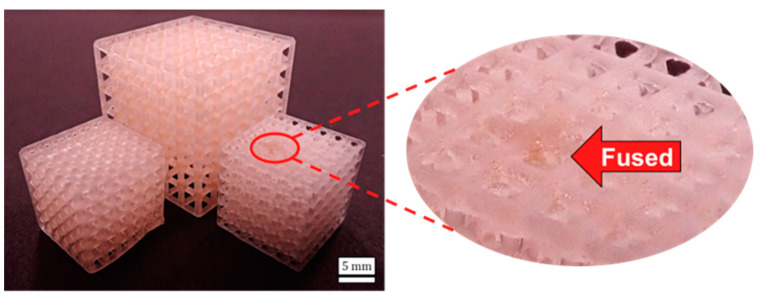
Fabricated lattices for Cube-800ø, BCC-800ø, and BCC-500ø designs (from **left** to **right**). Circled zoomed region indicates fused pores.

**Figure 7 polymers-14-05515-f007:**
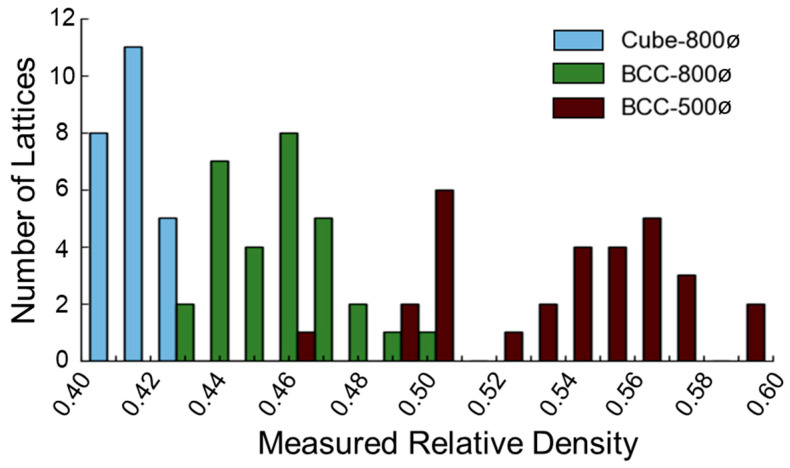
Histogram of measured relative density for thirty samples of each lattice design.

**Figure 8 polymers-14-05515-f008:**
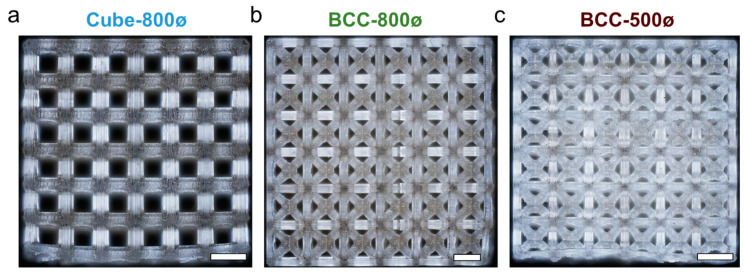
Microscopy of lattice side face for a representative sample of (**a**) Cube-800ø, (**b**) BCC-800ø, and (**c**) BCC-500ø designs. Scale bars approximately 2 mm.

**Figure 9 polymers-14-05515-f009:**
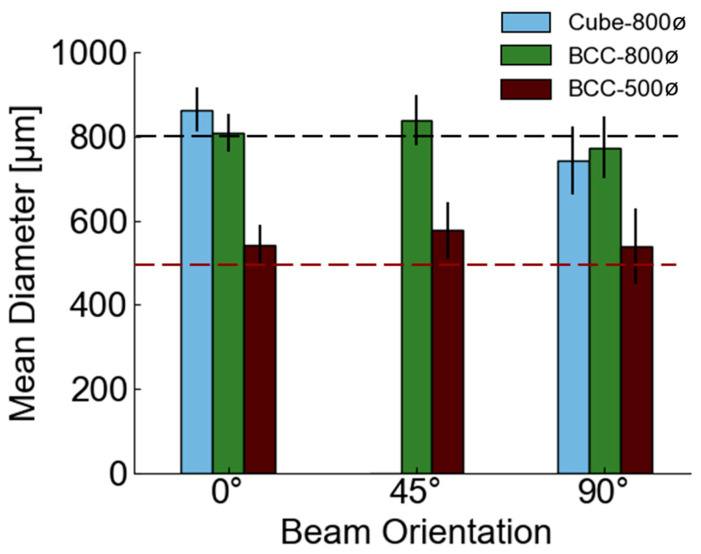
Measured mean beam diameters with standard deviation bars. Black dotted line provides a reference for 800 µm intended beam diameters, while the lower red dotted line provides a reference for 500 µm intended beam diameters.

**Figure 10 polymers-14-05515-f010:**
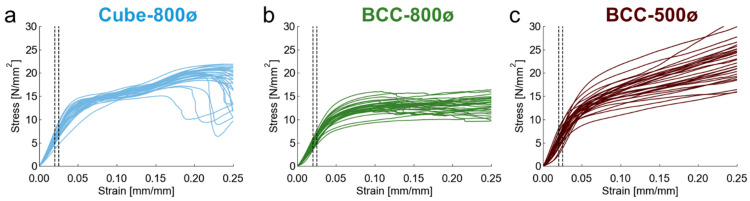
Stress-strain curves for (**a**) Cube-800ø, (**b**) BCC-800ø, and (**c**) BCC-500ø lattice designs for thirty samples each. Dotted vertical lines indicate boundaries for calculating elastic modulus for each curve.

**Figure 11 polymers-14-05515-f011:**
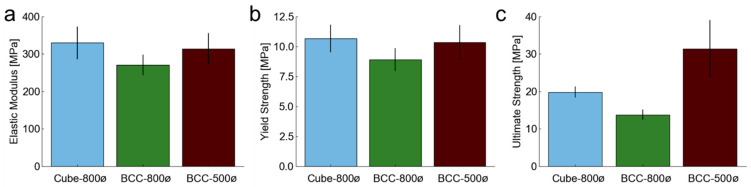
Mean mechanical property measurements for each lattice design with standard deviation bars aggregated from thirty samples for (**a**) elastic modulus *E_m_*, (**b**) yield strength *Y_s_*, and (**c**) ultimate strength *U_s_*.

**Figure 12 polymers-14-05515-f012:**
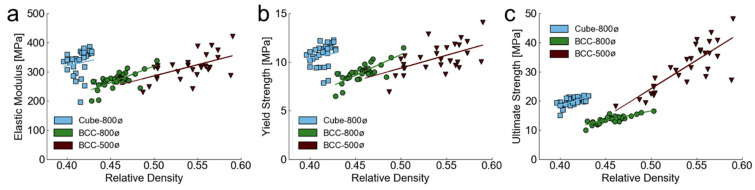
Mechanical property measurements for each tested sample according to relative density for (**a**) elastic modulus *E_m_*, (**b**) yield strength *Y_s_*, and (**c**) ultimate strength *U_s_*.

**Figure 13 polymers-14-05515-f013:**
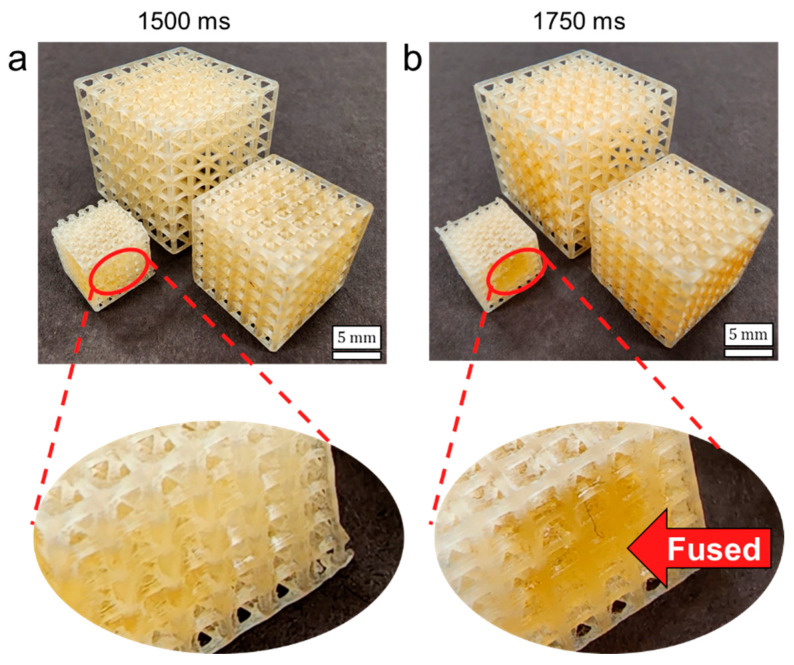
Fabricated BCC-0.5$\rho_{r}$, BCC-0.4$\rho_{r}$, and BCC-0.3$\rho_{r}$ lattices processed with exposure times per layer of (**a**) 1500 ms and (**b**) 1750 ms. Circled areas highlight the higher proportion of fused pores for the 1750 ms condition.

**Figure 14 polymers-14-05515-f014:**
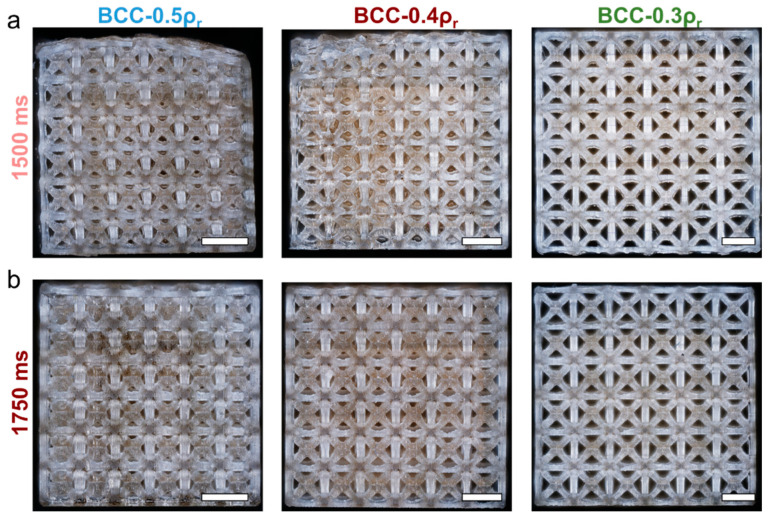
Microscopy of lattice side face for representative samples constructed with (**a**) 1500 ms and (**b**) 1750 ms exposure time per layer. Scale bars approximately 2 mm.

**Figure 15 polymers-14-05515-f015:**
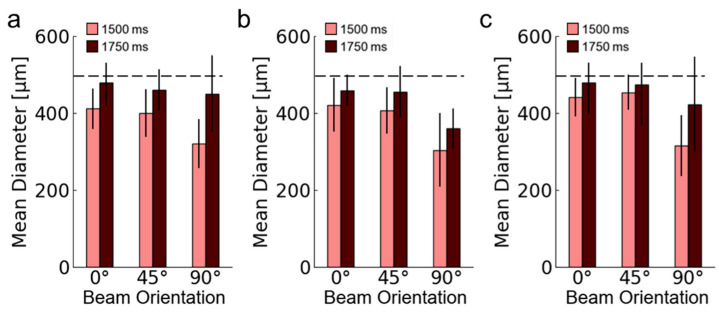
Measured mean beam diameters for 1500 ms and 1750 ms exposure per layer with standard deviation bars for designs of (**a**) BCC-0.5$\rho_{r}$, (**b**) BCC-0.4$\rho_{r}$, and (**c**) BCC-0.3$\rho_{r}$. Dotted lines provide references for the 500 µm intended beam diameter for each design.

**Figure 16 polymers-14-05515-f016:**
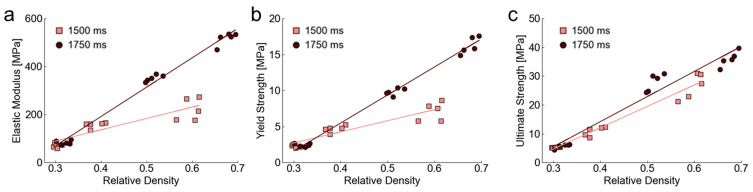
Mechanical property measurements for each tested sample of differing exposure times according to relative density for (**a**) elastic modulus *E_m_*, (**b**) yield strength *Y_s_*, and (**c**) ultimate strength *U_s_*.

**Table 1 polymers-14-05515-t001:** Design and mean measured dimensional characteristics of lattices. All lattices were designed with 40% relative density.

Design					Mean Measurements (± Standard Deviation)		
Name	Unit Cell Topology	Unit Cell Length [mm]	Beam Diameter [µm]	Lattice Height/Length [mm]	Lattice Height [mm]	Lattice Length [mm]	Relative Density
Cube-800ø	Cube	1.8	800	11.6	11.8 ± 0.06	11.8 ± 0.05	0.42 ± 0.01
BCC-800ø	BCC	3.1	800	19.4	19.5 ± 0.10	19.6 ± 0.09	0.46 ± 0.02
BCC-500ø	BCC	1.9	500	11.9	11.9 ± 0.07	12.0 ± 0.10	0.54 ± 0.03

**Table 2 polymers-14-05515-t002:** Linear regression values with correlation coefficient for mechanical properties of each design, according to Equation (3).

Design	Property *p**	*m*	*c*	R^2^
	*E_m_* *Y_s_* *U_s_*	641	64.0	0.02
Cube-800ø		19.4	2.6	0.02
		98.1	−20.8	0.39
	*E_m_* *Y_s_* *U_s_*	1116	−239.2	0.49
BCC-800ø		42.9	−10.7	0.60
		66.0	−16.4	0.70
	*E_m_* *Y_s_* *U_s_*	765	−96.7	0.33
BCC-500ø		26.3	−3.8	0.34
		194.6	−73.1	0.66

**Table 3 polymers-14-05515-t003:** Design and mean measured properties of lattices. All lattices have BCC topology with 500 µm beam diameters.

Design			Process	Mean Measurements (± Standard Deviation)			
Name	Relative Density	Unit Cell Length [mm]	Layer Exposure Time [ms]	Relative Density	Elastic Modulus [MPa]	Yield Strength [MPa]	Ultimate Strength [MPa]
$\mathrm{BCC}-0.5\rho_{r}$	0.5	2.3	1500	0.60 ± 0.02	222 ± 38	7.1 ± 1.1	27 ± 3.6
$\mathrm{BCC}-0.4\rho_{r}$	0.4	1.9	1500	0.39 ± 0.02	157 ± 10	4.7 ± 0.4	11 ± 1.3
$\mathrm{BCC}-0.3\rho_{r}$	0.3	1.6	1500	0.30 ± 0.01	71 ± 8	2.3 ± 0.2	5 ± 0.1
$\mathrm{BCC}-0.5\rho_{r}$	0.5	2.3	1750	0.68 ± 0.01	517 ± 22	16.3 ± 0.1	36 ± 2.2
$\mathrm{BCC}-0.4\rho_{r}$	0.4	1.9	1750	0.51 ± 0.01	351 ± 12	9.8 ± 0.4	28 ± 2.5
$\mathrm{BCC}-0.3\rho_{r}$	0.3	1.6	1750	0.32 ± 0.01	84 ± 7	2.4 ± 0.2	6 ± 0.6

**Table 4 polymers-14-05515-t004:** Linear regression values with correlation coefficient for mechanical properties to measured relative density of BCC lattices with varied layer exposure time, according to Equation (3).

Layer Exposure Time	Property *p**	*m*	*c*	R^2^
	*E_m_* *Y_s_* *U_s_*	471	−52.3	0.78
1500 ms		15.5	−1.9	0.85
		73.9	−17.5	0.97
	*E_m_* *Y_s_* *U_s_*	1222	−297	0.99
1750 ms		38.9	−10.1	0.99
		86.9	−20.5	0.95

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
