# Peer review of "Mechanics of 3D-Printed Polymer Lattices with Varied Design and Processing Strategies"

_polymers, 2022, doi:10.3390/polym14245515_

Round 1
Reviewer 1 Report
1. Please also use different patterns in the bars in Figures 7, 9, 11, and 15. Otherwise, it's difficult to distinguish the differences when the paper was printed in black and white.
2. In figures, the font size should be consistent, e.g., Figure 7.
3. Any physical meaning of Eq. 3?
Any future work?
Reviewer 2 Report
Please refer to the attached report

Round 2
Reviewer 1 Report
The authors have already modified the manuscript based on my comments. So, I believe it's ready to be published.
Reviewer 2 Report
The manuscript has been updated accordingly and ready to be accepted for publish. However, there is very minor correction need to be made before publish as below;
i. All figures need to label (a), (b), (c) etc and not a, b, c &
ii. Please use standard size font for labelling